# Breakup of the proton halo nucleus $^8$B near barrier energies

L. Yang[1,2], C. J. Lin [1,3] ✉, H. Yamaguchi [2,4], A. M. Moro [5,6] ✉, N. R. Ma[1,2], D. X. Wang[1], K. J. Cook [7,8,21], M. Mazzocco [9,10], P. W. Wen[1], S. Hayakawa [2], J. S. Wang [11], Y. Y. Yang[12], G. L. Zhang[13], Z. Huang[13], A. Inoue[14], H. M. Jia[1], D. Kahl [15], A. Kim[16], M. S. Kwag[17], M. La Commara[18], G. M. Gu[17], S. Okamoto[19], C. Parascandolo[20], D. Pierroutsakou[20], H. Shimizu[2], H. H. Sun[1], M. L. Wang[13], F. Yang[1] & F. P. Zhong[1,3]

The dynamics of a nuclear open quantum system could be revealed in the correlations between the breakup fragments of halo nuclei. The breakup mechanism of a proton halo nuclear system is of particular interest as the Coulomb polarization may play an important role, which, however, remains an open question. Here we use a highly efficient silicon detector array and measure the correlations between the breakup fragments of $^8$B incident on $^{120}$Sn at near-barrier energies. The energy and angular correlations can be explained by a fully quantum mechanical method based on the state-of-the-art continuum discretized coupled channel calculations. The results indicate that, compared to the neutron halo nucleus $^6$He, $^8$B presents distinctive reaction dynamics: the dominance of the elastic breakup. This breakup occurs mainly via the short-lived continuum states, almost exhausts the $^7$Be yield, indicating the effect of Coulomb polarization on the proton halo state. The correlation information reveals that the prompt breakup mechanism dominates, occurring predominantly on the outgoing trajectory. We also show that, as a large environment, the continuum of $^8$B breakup may not significantly influence elastic scattering and complete fusion.

The physics of open quantum systems (OQS), which is related with universality features and generic exotic quantum phenomena, has become one of the most intriguing topics in modern physics. An OQS may be defined as a system which is found to be in interaction with an external system, the environment. In the case of nuclear physics, the light nuclei far from the valley of beta stability are characterized by extremely low binding energies and extended nuclear matter distributions, leading in some cases to the formation of an exotic halo structure[1]. Such nuclei are hence excellent examples of a many-body OQS. Couplings to the continuum states play a significant role in the structure and reactions of these systems, a manifestation of which is the large breakup probability in nuclear collisions. The continuum hence provides a large environment, which strongly interacts with the subsystems, like, the elastic scattering channel related to the ground state[2]. Moreover, the influence of the continuum on the fusion reaction, especially at energies around the Coulomb barrier, is the frontier of fusion studies[3]. In this respect, studies of the breakup mechanism of a halo nuclear system open up a path for extended exploration of the dynamics of a nuclear OQS[4,5].

Complementary information of the breakup dynamics comes from correlations in relative motions between the breakup fragments[6–9]. For neutron-halo systems, owing to the difficulties of effective detection of neutrons, a few attempts have been made to measure the alphas and neutrons in coincidence for the $^6$He+$^{209}$Bi system[10]. The results established that 2$n$−transfer, rather than breakup, was the dominant direct process. For a proton halo nucleus, however,

the Coulomb barrier between the core and the valence proton prevents it from reaching the state of an OQS[5]. Moreover, the dynamic Coulomb polarization effect[11,12] may produce a hindrance of both the proton transfer and breakup processes[13]. Compared to their neutron counterparts, these distinctive properties result in the rather elusive character of proton halo systems. The proton dripline nucleus $^8$B has a very small proton separation energy of merely 138 keV, and it is one of the few observed cases where the ground state presents a proton halo, which could be indicated by the measurements of proton elastic scattering[14] and longitudinal momentum distributions of $^7$Be at high energies[15–18]. Hence, $^8$B provides an excellent candidate to investigate the breakup dynamics of proton halo nuclei. The breakup process of $^8$B is schematically depicted in Fig. 1.

Exclusive breakup measurements, namely breakup fragments measured in coincidence, have been achieved at high beam energies (40−80 MeV per nucleon) to investigate the Coulomb dissociation of $^8$B[19–21]. Multiple reaction models[22–27] were developed as well to study the breakup process at high energies. However, it becomes more challenging for incident energies close to the Coulomb barrier, since it is not easy to carry out a coincidence measurement between the breakup fragments as was done at higher energies, due to the much reduced kinematic focusing. Despite the substantial efforts devoted[28–31], the low beam intensity and insufficient detection efficiency have prevented so far coincident measurements of the $^8$B system. To date, a few inclusive breakup measurements (only one of the breakup fragments is recorded), namely, $^8$B+$^{58}$Ni[28], $^{64}$Zn[31], and $^{208}$Pb[30], have been reported at energies around the Coulomb barrier. The inclusive angular distributions of the $^7$Be core are consistent with continuum discretized coupled channel (CDCC) calculations[30–33], indicating the predominance of the elastic breakup (EBU), i.e., all of the outgoing particles are emitted in their ground states. Experiments are desired to verify this conclusion. Moreover, the elastic scattering measurements of $^8$B+$^{12}$C[34], $^{27}$Al[35], $^{58}$Ni[36], $^{64}$Zn[31], and $^{208}$Pb[29] demonstrate modest coupling effects to the continuum. By contrast, for fusion reactions, contradictory conclusions were obtained from the measurements of $^8$B+$^{28}$Si[37], $^{40}$Ar[38], and $^{58}$Ni[39]: striking enhancement was reported for $^8$B+$^{58}$Ni, which was not observed in $^8$B+$^{28}$Si and $^{40}$Ar. The breakup mechanism is essential to understand this inconsistency, since, if the breakup occurs on the incoming trajectory, it might cause a loss of flux for fusion and suppress the cross section. Due to the lack of coincident measurements, however, the detailed breakup mechanism of a proton-halo nuclear system remains an open question.

In this work, we present the results of the complete kinematics measurement of $^8$B+$^{120}$Sn at two energies around the Coulomb barrier to elucidate the breakup mechanism of $^8$B. We choose $^{120}$Sn as the target, since the maximum available $^8$B beam energy could match with its Coulomb barrier height, which enables us to investigate the Coulomb polarization effect and the Coulomb-nuclear interference as well. Using a detector array with large solid-angle coverage, we derived the correlation between the breakup fragments, $^7$Be and $p$. The breakup intermediate states are reconstructed accordingly and a comprehensive breakup picture is derived successfully. The results show that the prompt breakup mechanism dominates, occurring predominantly on the outgoing trajectory.

## Results and discussion

The experiment was performed at the low-energy radioactive-ion beam facility CNS Radioactive Ion Beam separator (CRIB)[40,41] of the Center for Nuclear Study (CNS), the University of Tokyo. The secondary $^8$B radioactive beam with a typical intensity of $1 \times 10^4$ particles per second and a purity of ~20%, impinged onto a 2.7 mg cm$^{-2}$-thick, self-supporting $^{120}$Sn secondary target (97% isotopically enriched). By inserting an aluminum degrader after the primary target, we obtained $^8$B beams with two distinct energies of 38.7 ± 0.5 and 46.1 ± 0.6 MeV in the middle of the $^{120}$Sn target. Two parallel plate avalanche counters

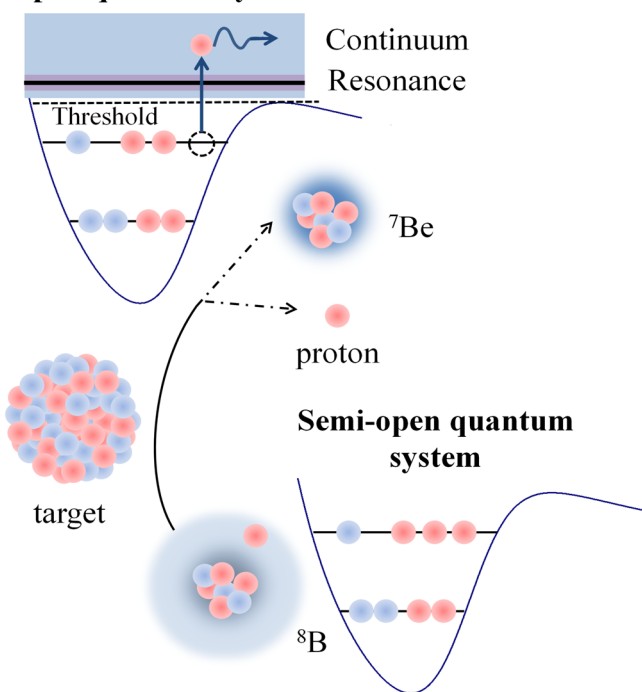

**Open quantum system**

**Fig. 1 | Illustration of $^8$B breaking up into $^7$Be and proton.** In the initial state, the nucleons in $^8$B occupy bound single particle orbits in the potential well. Although located in the potential well, the halo proton is extremely weakly-bound in an extended distribution around the centrum $^7$Be core. It hence can be regarded as a semi-open quantum system. During the collision, $^8$B is excited to the unbound states above the breakup threshold (either resonant or non-resonant), forming an open quantum system.

(PPACs)[42] were installed in front of the target to reconstruct the trajectory of each incident beam ion event by event. A highly efficient silicon detector array[43] was employed to detect the reaction products. A summary on the experiment is given in the "Methods" section (see below).

## Angular distributions

The angular distributions of elastic scattering relative to Rutherford scattering at 38.7 and 46.1 MeV are shown, respectively, in Fig. 2a, b by squares, where only the statistical uncertainties are considered. The scattering data were normalized to a Monte Carlo simulation[44], which assumes a pure Rutherford scattering at all angles and considers the geometry of the detector array and the beam position determined by the PPACs. To interpret the scattering data, CDCC calculations were performed using the codes THOx[45] and FRESCO[46]. In Fig. 2, the elastic scattering data are compared with the CDCC results (solid curves). Overall, the agreement is very satisfactory. We include also the "one-channel" calculations (dashed curves) in which the couplings to the continuum are switched off. One can see that, although it is not as significant as in the neutron halo systems[47,48], the influence of the continuum states on the elastic scattering cannot be neglected, showing a suppression of the Coulomb-nuclear interference peak. The total reaction cross section derived from the elastic scattering data are 874.1 and 1264.0 mb for 38.7 and 46.1 MeV, respectively.

The angular distributions of inclusive and exclusive $^7$Be produced from $^8$B+$^{120}$Sn reactions are shown in Fig. 2 as well by diamonds and stars, respectively. One can see that the inclusive and exclusive results are almost identical to each other within the uncertainties, providing a clear experimental evidence that breakup, rather than the proton transfer, dominates the $^7$Be production mechanism in the reaction of the proton halo $^8$B system. This is opposed to the situation for the

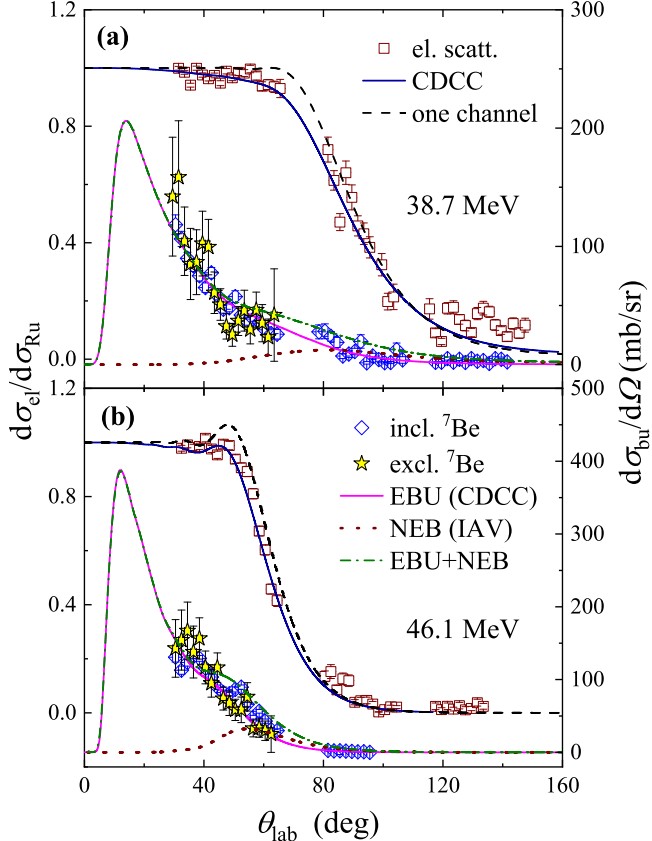

**Fig. 2 | Angular distributions of elastic scattering and breakup reactions.** Squares, diamonds and stars denote the experimental data of elastic scattering, inclusive and exclusive breakup at **a** 38.7 and **b** 46.1 MeV, respectively. The elastic scattering and breakup data are respectively related to the left and right axes. The error bars indicate the statistical uncertainties. CDCC results for elastic scattering and elastic breakup (EBU) are shown by the blue and magenta solid curves, respectively. As a comparison, the one-channel calculations for the elastic scattering are represented by the dashed curves. The dotted lines correspond to the non-elastic breakup (NEB) contributions, which are derived from the IAV model calculations. The dash-dotted lines stand for the sum of EBU and NEB.

neutron halo nucleus $^6$He, for which the neutron transfer dominates according to the coincident measurement of ref. 10. In semiclassical terms, Coulomb polarization favors neutrons in the halo residing in the region between the core and the target, which then enhances the transfer probabilities. It is similar to the so-called Oppenheimer-Phillips process[49], a manifestation of which is that the cross section of (d, p) is enhanced as compared to (d, n) cross section owing to the strong electric polarizability of the weakly bound deuteron. For the case of the proton halo nucleus $^8$B, the Coulomb polarization would result in the valence proton being displaced behind the core leading to a decoupling between the core and the valence proton[50]. In this sense, the breakup cross section of a proton halo nucleus is expected to be large.

Two types of non-elastic breakup (NEB) processes might be included in the exclusive data: the core ($^7$Be) and the target excitations. Due to the energy straggling of the $^8$B beam and the limitation of statistics, these excited states cannot be resolved from the ground state experimentally. The $^7$Be system has one bound excited state ($E_x = 0.43$ MeV, $J^\pi = 1/2^-$) below the breakup threshold. As shown in Fig. 2 by the solid curves, even though not included the $^7$Be excitation, CDCC calculations describe both the elastic scattering and exclusive breakup data fairly well, indicating that the contribution of the core excitation may not be significant. As a comparison, for the neutron halo nuclear system $^{11}$Be+$^{197}$Au[47], the elastic

scattering data could only be described when the core-excited admixtures are taken into account. On the other hand, extended CDCC calculations based on the formalism of ref. 51 were performed to estimate the contribution of $^{120}$Sn target excitation to the first excited state ($E_x = 1.17$ MeV, $J^\pi = 2^+$) on the breakup cross sections. The results indicate that this contribution is very small: 2.4 and 2.7 mb for 38.7 and 46.1 MeV, respectively. This clearly corroborates the dominance of the elastic breakup mechanism on the measured coincidence data.

Furthermore, calculations based on the Ichimura–Austern–Vincent (IAV)[31,52] model were performed to evaluate the contributions to the inclusive $^7$Be yield from the NEB processes, which include non-capture breakup accompanied by target excitation, proton absorption by the target (incomplete fusion) and proton transfer leading to bound states of $^{121}$Sb. The NEB calculations results as well as the sum of EBU and NEB are displayed in Fig. 2 by the dotted and dash-dotted curves, respectively. One can see that the inclusive breakup data agree well with the sum of EBU and NEB. The calculated cross-sections of EBU and NEB are, respectively, $\sigma_{EBU} = 351.5$ (420.5) mb and $\sigma_{NEB} = 78.3$ (91.4) mb for 38.7 (46.1) MeV. It demonstrates clearly that NEB contribution, although not negligible (-18% of the total $^7$Be yield), is just minor.

### Correlations between the breakup fragments

Correlations between the breakup fragments provide key information to pin down the underlying breakup dynamics. In particular, the relative energy $E_{rel}$ between the breakup fragments is a critical quantity to infer the projectile excitation and location of breakup[6,8,53]. $E_{rel}$ is defined as:

$$E_{rel} = \frac{m_1 E_2 + m_2 E_1 - 2\sqrt{m_1 E_1 m_2 E_2} \cos\theta_{12}}{m_1 + m_2}, \tag{1}$$

where, $m_1$, $m_2$ and $E_1$, $E_2$ are the deduced masses and energies of the breakup fragments and $\theta_{12}$ their opening angles. The reconstructed $E_{rel}$ distributions (without efficiency correction) of $^7$Be+$p$ at 38.7 and 46.1 MeV are presented with the circles in Fig. 3a and b, respectively, where the solid curves denote the $E_{rel}$ distributions reconstructed with the simulation described in the "Methods" section. In the simulation, the detector geometry, pixelation[54] and energy resolution were incorporated explicitly. Overall, the simulations agree well with the experimental data at both energies, although some deviations are apparent at 46.1 MeV. This might be due to the choice of the $^7$Be and proton+$^{120}$Sn potentials, which are derived from global parameterizations. The effect of these potentials will be more important at 46.1 MeV because it is more above the Coulomb barrier. A peak at around 0.6 MeV is observed in Fig. 3. This peak is very close to the first resonance of $^8$B ($E_x = 0.77$ MeV, $J^\pi = 1^+$, $\Gamma = 35.6$ keV), the position of which in $E_{rel}$ is indicated by the vertical line. Owing to the narrow width, this $1^+$ resonance might be the only state that can be observed as a distinctive peak in the $E_{rel}$ distribution. To highlight its relative contribution, we show in the insets of Fig. 3 the calculated $E_{rel}$ distributions for the calculation with the orbital angular momentum up to $l = 3$ (solid lines) and for the $p$-wave $1^+$ state (dashed curves). The simulated $E_{rel}$ distributions via the $p$-wave $1^+$ state are shown in Fig. 3 by the dashed curves. By integrating the peak region, the contributions from the $1^+$ resonance are determined as $(4.4 \pm 2.0)\%$ and $(3.8 \pm 2.5)\%$ at 38.7 and 46.1 MeV, respectively. The lifetime of this resonance is about $10^{-20}$ s. According to the result of $^8$Li[8], such a timescale is sufficiently long that breakup via this state mainly occurs on the outgoing trajectory, receding from the target. The small fraction of the $1^+$ resonance compared to non-resonant contributions suggests nevertheless that the prompt component dominates the breakup mechanism.

Additional information on the timescale of breakup can be obtained from the distribution of $\theta_{12}$ and the orientation of the relative momentum of the breakup fragments ($\beta$) in their center-of-mass

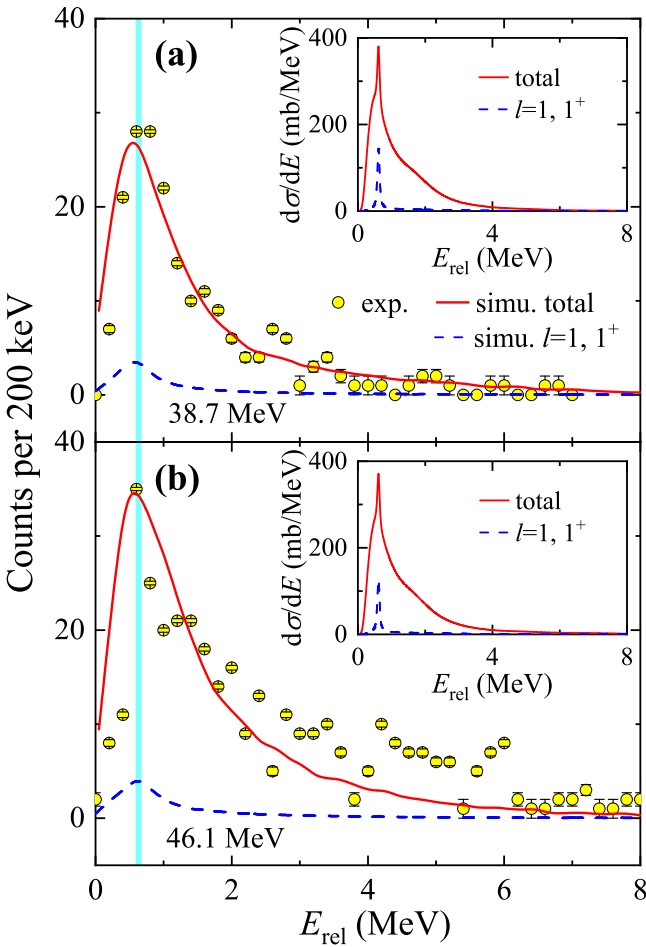

**Fig. 3 | Measured $E_{rel}$ distributions for breakup fragments $^7$Be and proton.** The experimental data (circles) at **a** 38.7 and **b** 46.1 MeV are compared with the simulated distributions of $E_{rel}$ (solid curves). The error bars show the associated statistical uncertainties. The dashed curves denote simulation results of $p$-wave $1^+$ state. The raw theoretical $E_{rel}$ distributions from CDCC are shown in the insets, where the solid and dashed curves represent the calculations with the orbital angular momentum up to $l = 3$ (total) and the contributions from the $p$-wave ($l = 1$) $1^+$ state. The vertical line indicates the expected location of the peak from the first $1^+$ resonance of $^8$B.

frame[53–55]. The angle $\beta$ is determined from

$$\sin\beta = \frac{v_1 v_2 \sin\theta_{12}}{(v_2^2 u_1^2 + v_1^2 u_2^2 + 2u_1 u_2 v_1 v_2 \cos\theta_{12})^{1/2}}, \quad (2)$$

where $v_i$ and $u_i$ are the particle velocities in the laboratory and composite rest frames, respectively. The definitions of $\theta_{12}$ and $\beta$, as well as the corresponding velocities are depicted in the inset of Fig. 4a. The angles $\theta_{12}$ and $\beta$ are highly correlated when breakup occurs asymptotically with a well-defined excitation energy. The measured $\theta_{12}$-$\beta$ distributions are shown in Fig. 4a, d. The dashed lines overlaid on the data correspond to the expected correlation between $\theta_{12}$ and $\beta$[53] for asymptotic breakup from the $1^+$ resonant state of $^8$B. The events following the dashed lines are identified with breakup products which are weakly affected by the presence of the target Coulomb field and therefore they presumably break up far from the target. Conversely, for events arising from breakup near the target, the $\theta_{12}$-$\beta$ correlation will be distorted due to the Coulomb postacceleration[53]. As shown in Fig. 4a, d, a majority of the events deviate from the asymptotic limit, further confirming that the near-target component is the dominant breakup mode in the present reaction.

The simulations based on the CDCC calculations reasonably reproduce the experimental $\theta_{12}$-$\beta$ distributions as shown in Fig. 4a, d. The projections of $\beta$ and $\theta_{12}$ at 38.7 MeV are shown in Fig. 4b, c, and the results at 46.1 MeV are presented in Fig. 4e, f, where the solid curves represent the simulations. For $\theta_{12}$, the bulk of the events appears at low angles, which peaks around 30°, and the simulation qualitatively reproduces the experimental data. It has been shown that the $\theta_{12}$ distribution gives information on the location of breakup[53]: When the projectile breaks up on the incoming trajectory, the initial velocity of the breakup fragments is towards the target nucleus, which is against the repulsive Coulomb field and results in a large $\theta_{12}$. By contrast, in the case of breakup along the outgoing trajectory, the initial velocities of the fragments are in the same direction as the Coulomb interaction from the target nucleus, leading to a small $\theta_{12}$. Therefore, the strongly forward-peaked $\theta_{12}$ distributions suggest that most of the breakup, while prompt, occurs along the outgoing trajectory, and hence with potentially insignificant influence on the complete fusion cross section of $^8$B.

The $\beta$ distribution is reconstructed assuming the breakup occurs far from the target. If breakup occurs close to the target, the reconstructed $\beta$ will be distorted by the post-breakup fragment-target interactions. Therefore, owing to the dominance of the near-target breakup, the distribution of $\beta$ will not reflect the initial orientation of $^8$B*. Although the postacceleration effect is taken into account by the CDCC calculations, the simulation is found to underestimate the experimental data around 90°. This discrepancy might indicate some inadequacy of the $^7$Be+$^{120}$Sn interaction parameters and/or the simplified $^8$B structure model adopted in the CDCC calculation, which would deserve further investigation.

Before closing this section, we would like to add a brief comment about the halo property of the $^8$B nucleus as reflected in the longitudinal momentum distribution of the $^7$Be fragment. Indeed, it has been found that the narrow longitudinal momentum distribution of the $^7$Be fragment from breakup reactions is regarded as prominent evidence of a halo structure of $^8$B. Therefore, we extracted the corresponding results from the present coincident data. Gaussian-like structures are observed at both energies, and the full widths at half maximum (FWHM) are determined as $88 \pm 22$ and $106 \pm 14$ MeV/c for 38.7 and 46.1 MeV, respectively. These results are comparable with those derived with high-energy $^8$B beams, like, $93 \pm 7$ MeV/c at 38 MeV/nucleon[17], $85 \pm 4$ MeV/c at 41 MeV/nucleon[16], $81 \pm 6$ MeV/c at 1471 MeV/nucleon[15], and $92 \pm 7$ MeV/c at 36 MeV/nucleon[18]. It is worth noting that the present result extracted from low energies includes the influence from couplings, Coulomb post-acceleration and beam energy straggling. These effects, however, are still difficult to be precisely estimated.

The breakup dynamics of a proton halo nucleus is a long-standing question. Using a highly efficient detector array, the correlation between the breakup fragments of $^8$B colliding with a $^{120}$Sn target was obtained at energies close to the Coulomb barrier. The results show that the yield of $^7$Be is almost exhausted by the elastic breakup, which occurs mainly via the short-lived continuum states, on the outgoing trajectory. Compared with the neutron halo system, the present results demonstrate distinctive OQS dynamics of a proton-halo nucleus, which is mainly due to the Coulombian effect of the halo proton. This breakup picture indicates that, although it provides a large environment, the continuum has minor influence on elastic scattering and complete fusion. To further elucidate the breakup behavior of a proton halo nuclear system, a detailed and quantitative analysis on the orientation effect is desired. This work demonstrates the applicability of the complete kinematics measurement combining the fully quantum analysis framework. This technique will be applied to further understanding the dynamics of nuclear OQS via the breakup reaction.

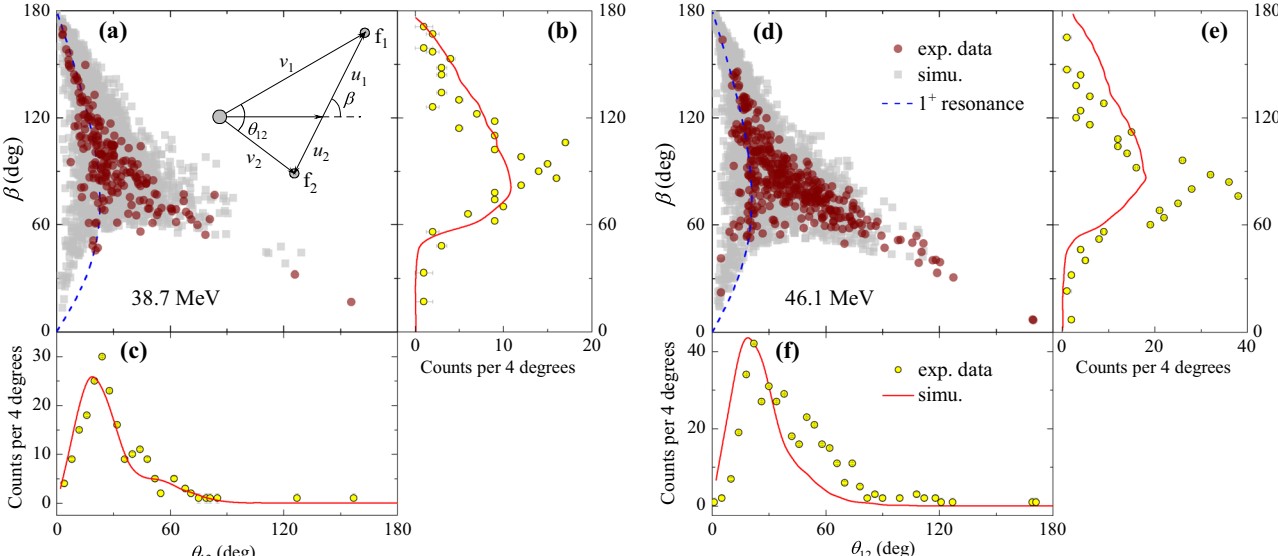

**Fig. 4 | Angular correlations of breakup fragments $^7$Be and proton.** Comparison of experimental data (circles, with error bars indicating the associated statistical uncertainties) with simulations (squares) for correlations of $\beta$ and $\theta_{12}$ at **a** 38.7 and **d** 46.1 MeV. The dashed curves show the expected $\beta$-$\theta_{12}$ correlation assuming asymptotic breakup from the 1$^+$ resonance of $^8$B. The projections of $\beta$ and $\theta_{12}$ at the corresponding energies are shown in **b**, **c**, and **e**, **f**, respectively, where the solid curves represent the simulations based on the CDCC calculations. The inset in **a** illustrates the definitions of $\beta$ and $\theta_{12}$. The error bars show the associated statistical uncertainties.

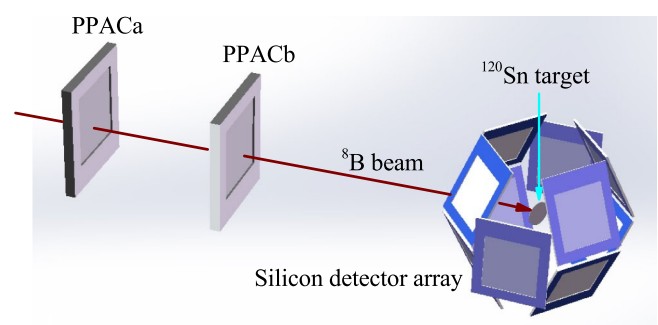

**Fig. 5 | Schematic view of the experimental setup.** The upstream and downstream parallel plate avalanche counters (PPACs) are denoted as the PPACa and PPACb, respectively. Ten silicon detector telescopes surround the $^{120}$Sn target, arranged in a spherical shape with a radius around 70 mm. To demonstrate clearly, only the first layers of the silicon detector array are displayed.

## Methods

### Experiment and data analysis

The $^8$B radioactive beam was produced via the $^3$He($^6$Li,$^8$B) reaction in inverse kinematics by using a 11.2 MeV/nucleon $^6$Li primary beam accelerated by the RIKEN AVF cyclotron. A cryogenic $^3$He gas target[56] was used as the primary target. $^7$Be and $^3$He were the main contaminants in the secondary beam.

The experimental setup is presented in Fig. 5. Two position-sensitive PPACs were installed in front of the $^{120}$Sn target. The distances from the upstream PPAC (PPACa) to the downstream PPAC (PPACb) and to the $^{120}$Sn target are 248 and 544 mm, respectively. The trajectory of each incident beam ion was reconstructed event by event according to the time difference between the signals arriving at the ends of the delay lines of the PPAC cathodes. The reaction products were detected by a highly efficient silicon-detector array, which comprises ten independent telescopes, arranged in a spherical shape with a radius around 70 mm. Each telescope consists of three stages of silicon detectors: the first layer is one 40/60 μm double-sided silicon strip detector (DSSD) with an effective area of $50 \times 50$ mm$^2$, followed by two layers of quadrant silicon detectors with thicknesses of 1000/1500 μm. Thanks to such a compact structure, the array covered about 40% of the total solid angle with an angular coverage of 24$^\circ$–158$^\circ$. A collimator with a diameter of 20 mm was installed at the entrance of the silicon detector array (116 mm to the target) to confine the beam spot profile. The data acquisition (DAQ) system trigger condition employed was the AND of the anode signal of the upstream PPAC and the OR of the silicon detectors. This trigger condition was used for both the inclusive and exclusive measurements. Trigger rates of about 1 kHz were typically recorded during the experiment. Besides the DAQ for physical runs, another independent DAQ (CRIB-DAQ) was used for the beam tuning and PPAC on-line monitoring. For this CRIB-DAQ, the down-scaled PPACa anode signal was used as the trigger, and the "pileup" circuit was introduced to reject the sequential signals from PPACa within 500 ns. With the data recorded by the CRIB-DAQ, the efficiency of the PPACs were around 95% during the beam time.

The detectors were calibrated in energy by means of the scattered beam particles of $^8$B and $^7$Be, as well as the standard α sources containing the following radioisotopes: $^{148}$Gd, $^{237}$Np, $^{241}$Am, and $^{244}$Cm. After the energy calibration, all of the DSSD pixels can be aligned in energy. A typical $\Delta E - E_r$ (residual energy) spectrum recorded by one strip of the forward-angle telescope is shown in Fig. 6a, where the loci of scattering events of the $^8$B and $^7$Be beams, as well as protons are clearly visible. The $^7$Be fragment resulting from $^8$B reactions was unambiguously separated from $^7$Be beam impurities using time-of-flight (TOF) techniques, as illustrated in Fig. 6b, since the $^7$Be from $^8$B breakup has the same TOF as the $^8$B beam. The TOF was obtained from the time difference between the occurrence of an $E$ signal in a telescope and the radio-frequency timing pulse from the cyclotron. To identify the coincident $^7$Be-$p$ events, besides setting the energy gates in the $\Delta E - E_r$ spectrum to select the $^7$Be and $p$ particles, we also set a timing gate, that is, for each selected breakup fragment $^7$Be, we search for the coincident $p$ within a timing window of 20 ns. The events hitting the inter-strip gap were removed by setting an energy gate that both sides of the DSSD have registered the same energy in each event within 150 keV. The fraction of these inter-strip events is estimated to be less than 5%. This efficiency loss has been considered in the Monte Carlo simulation.

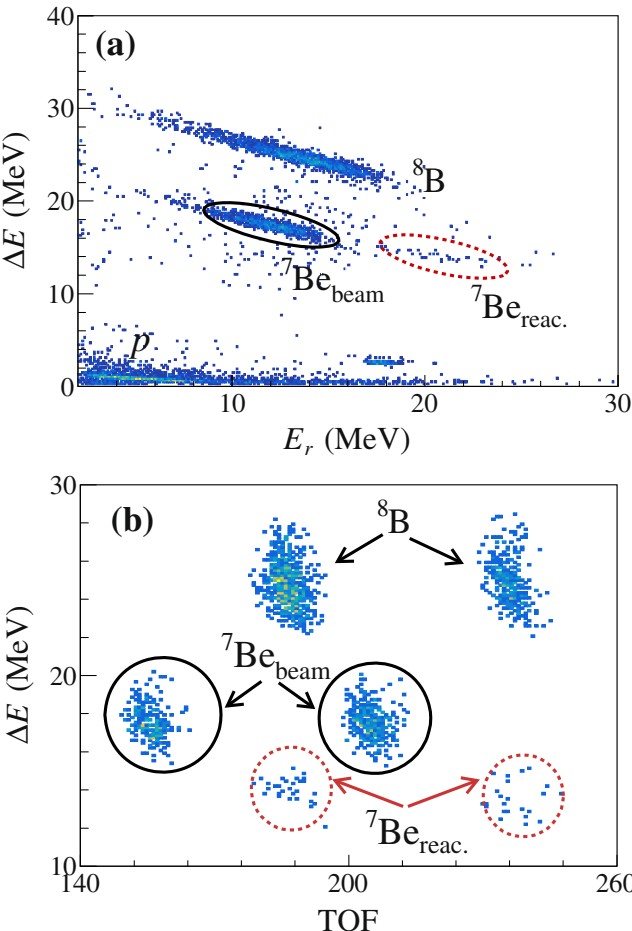

**Fig. 6 | Particle identification. a** Typical $\Delta E - E_r$ spectrum at 46.1 MeV, taken by one strip of the forward-angle telescope, covering the polar angular range of 30.4°–50.9°. **b** $\Delta E$-TOF spectrum of two radio-frequency cycles illustrating the separation between the $^7$Be from $^8$B reactions ($^7$Be$_{reac.}$) and elastically scattered $^7$Be in the secondary beam ($^7$Be$_{beam}$), as surrounded by the dashed and solid circles, respectively.

## CDCC framework

Standard CDCC calculations are employed to describe the elastic scattering and exclusive data in the present work, assuming a two-body model ($p+^7$Be) for the $^8$B projectile. Bound and unbound (continuum) wave functions of this system are generated with the THOx code[45] using the pseudo-state method, which consists in diagonalizing the projectile two-body Hamiltonian in a basis of square-integrable functions. In particular, we use the analytical Transformed Harmonic Oscillator (THO) basis[57,58], which is obtained by application of a local scaled transformation to the conventional harmonic oscillator basis. The calculated wave functions are then used to calculate the coupling potentials which, in turn, are used to solve the set of coupled differential equations with the coupled-channels code FRESCO[46]. The $p+^7$Be interaction contains central ($V_c(r)$) and spin-orbit ($V_{LS}(r)$ and $V_{Ll}(r)$) terms:

$$V(r) = V_c(r) + V_{LS}(r)\mathbf{l} \cdot \mathbf{s} + V_{Ll}(r)\mathbf{l} \cdot \mathbf{I}, \qquad (3)$$

where $\mathbf{l}$ is the $p$-$^7$Be relative orbital angular momentum, and $\mathbf{s}$ and $\mathbf{I}$ are the proton and $^7$Be spins, respectively. The radial dependence of each of these terms and their associated parameters can be found in ref. 31, where this model was successfully employed to analyze the $^8$B+$^{64}$Zn reaction. The model reproduces the position of the well-known low-lying $1^+$ and $3^+$ resonances at $E_x = 0.77$ and 2.32 MeV, respectively, and

the tentative $0^+$ resonance reported in ref. 59 at 1.90 MeV. It also predicts a $2^+$ resonance at 2.48 MeV, which might correspond to the $2^+$ resonance at 2.55 MeV proposed in the R-matrix analysis of ref. 59.

For the angular momentum part, the adopted coupling scheme is given by $|(ls)j, I; JM\rangle$, $J$ and $M$ the total $^8$B spin and its z-axis projection. Using this basis, and with the adopted model Hamiltonian, the $^8$B states include some configuration mixing. For example, the $^8$B ground state is a mixture of $p_{1/2}$ and $p_{3/2}$ configurations with weights 0.059 and 0.941, respectively, which are in reasonable agreement with ab-initio variational Monte Carlo spectroscopic factors 0.062(1) and 0.871(4)[60].

The calculation of the coupling potentials require also effective interactions (optical potentials) between the projectile subsystems and the target. For $^7$Be+$^{120}$Sn, since there are no experimental elastic scattering data of $^7$Be+$^{120}$Sn, we relied on a global $^7$Li potential of Cook[61], whereas the proton-$^{120}$Sn potential was taken from the global parametrization of Koning and Delaroche[62].

### Detection efficiency

For the exclusive breakup measurement, it is essential to determine the efficiency of detecting the breakup fragments in coincidence. To find this efficiency realistically, a novel simulation approach is established, based on the detailed outputs of CDCC. First, five-fold differential cross sections, $d\sigma/d\Omega_{^7Be}dE_{^7Be}d\Omega_p$, are produced from the CDCC breakup amplitudes[63]. The Markov chain Monte Carlo (MCMC) method[64], which is widely used in machine learning, is then employed to sample and obtain the $E_{^7Be}$, $\theta_{^7Be}$, $\theta_p$ and $\phi_p$ in the laboratory frame from the five-fold parameter space, assuming $\phi_{^7Be}$ has a flat distribution. The only unknown parameter, the proton energy $E_p$, can be determined based on the kinematics of the breakup reaction. Finally, the coincident detection efficiency is deduced from a Monte Carlo simulation with the complete kinematics information of each fragment and the detector array geometry.

## Data availability

The data that support the findings of this study are available from the corresponding author upon request.

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

## Acknowledgements

We are grateful to Alexis Diaz-Torres for a critical reading of the manuscript. This experiment was performed at RI Beam Factory operated by RIKEN Nishina Center and CNS, the University of Tokyo. This work is supported by the National Key R&D Program of China (Contract No. 2018YFA0404404), the National Natural Science Foundation of China (Grant Nos. U2167204, 12175314, 12235020, and U18672122), the Continuous Basic Scientific Research Project (No. WDJC-2019-13), the Leading Innovation Project (Grant nos. LC192209000701, LC202309000201), and the Basic Scientific Research Program under grant no. JCKY2020201C002. H.Y. is supported by JSPS KAKENHI (No. 19K03883). K.J.C. acknowledges the JSPS International Fellowship for Research in Japan, hosted by the Tokyo Institute of Technology. A.M.M. is supported by the project PID2020-114687GB-I00 (funded by MCIN/AEI/10.13039/501100011033), by the grant Group FQM-160 and by project P20_01247 (funded by the Consejería de Economía, Conocimiento, Empresas y Universidad, Junta de Andalucía, Spain and by "ERDF A way of making Europe"). M.S.K. and G.M.G. are supported by the National Research Foundation of Korea (NRF) grants funded by the Korea government (MSIT) (Grant No. 2013M7A1A1075764). A.K. acknowledges the National Research Foundation of Korea (NRF) grants funded by the Korea government (Grant No. 2018R1A5A1025563). J.S.W. is supported by the National Natural Science Foundation of China (Nos. U2032140 and 11875297).

## Author contributions

L.Y. prepared the proposal for the experiment, performed data analysis and wrote the manuscript. C.J.L. supervised the project. A.M.M. carried out the theoretical calculations. P.W.W. performed the MCMC samplings. L.Y., C.J.L., H.Y., N.R.M., D.X.W., K.J.C., M.M., S.H., J.S.W, Y.Y.Y., G.L.Z., Z.H., A.I., H.M.J., D.K., A.K., M.S.K., M.L.C., G.M.G., S.O., C.P., D.P., H.S., H.H.S., M.L.W., F.Y., and F.P.Z. performed the experiment. All authors discussed and commented on the manuscript.

## Competing interests

The authors declare no competing interests.

## Additional information

**Materials and Correspondence** Correspondence and material requests could be addressed to the first author, L. Yang (email: yang_lei@ciae.ac.cn).

[1]China Institute of Atomic Energy, Beijing, China. [2]Center for Nuclear Study, University of Tokyo, Wako, Saitama, Japan. [3]College of Physics and Technology & Guangxi Key Laboratory of Nuclear Physics and Technology, Guangxi Normal University, Guilin, China. [4]National Astronomical Observatory of Japan, Mitaka, Tokyo, Japan. [5]Departamento de FAMN, Universidad de Sevilla, Sevilla, Spain. [6]Instituto Interuniversitario Carlos I de Física Teórica y Computacional (iC1), Sevilla, Spain. [7]Department of Physics, Tokyo Institute of Technology, Meguro, Tokyo, Japan. [8]Facility for Rare Isotope Beams, Michigan State University, East Lansing, MI, USA. [9]Dipartimento di Fisica e Astronomia, Universita di Padova, Padova, Italy. [10]Istituto Nazionale di Fisica Nucleare-Sezione di Padova, Padova, Italy. [11]School of Science, Huzhou University, Huzhou, China. [12]Institute of Modern Physics, Chinese Academy of Sciences, Lanzhou, China. [13]School of Physics, Beihang University, Beijing, China. [14]Research Center for Nuclear Physics, Osaka University, Ibaraki, Japan. [15]Extreme Light Infrastructure – Nuclear Physics, Horia Hulubei National Institute for R& D in Physics and Nuclear Engineering (IFIN-HH), Bucharest-Măgurele, Romania. [16]Center for Extreme Nuclear Matters, Korea University, Seoul, Korea. [17]Department of Physics, Sungkyunkwan University, Suwon, Republic of Korea. [18]Department of Pharmacy, University Federico II, Napoli, Italy. [19]Department of Physics, Kyoto University, Kyoto, Japan. [20]Istituto Nazionale di Fisica Nucleare-Sezione di Napoli, Napoli, Italy. [21]Present address: Department of Nuclear Physics and Accelerator Applications, Research School of Physics, The Australian National University, Canberra, ACT, Australia. ✉e-mail: cjlin@ciae.ac.cn; moro@us.es

