## [Peer Review File · Nature Communications]

REVIEWER COMMENTS

Reviewer #1 (Remarks to the Author):

Dear Editor,

In this manuscript, Yang et al. study the breakup of ^8B into ^7Be and a proton at beam energies close to the Coulomb barrier. Very valuably, this study includes both the experimental measurement and a detailed theoretical analysis of the data.

The nucleus ^8B is an interesting system. First, being very proton rich and with a one-proton separation energy of less than 200keV, it is believed to host a one-proton halo in its one and only bound state. Second, the one-proton radiative capture that leads to its synthesis within star $^7\text{Be}(p,\gamma)^8\text{B}$ has played a crucial role in the solar-neutrino problem. The neutrinos detected by Ray Davis and his collaborators came mostly from the beta decay of ^8B into two alphas. Coulomb breakup, which can be seen as the time-reversed reaction of this radiative capture, has thus been used to constrain experimentally the reaction rate at stellar energies.

Albeit intriguing, the structure of ^8B remains elusive to nuclear physicists because of the difficulty to produce it as a beam. As noted by the authors in their introduction, various experiments have been performed to infer information about its halo structure and/or constrain the radiative-capture cross section within the Gamow peak. Unfortunately, because of the low statistics, most of the breakup reactions measured with ^8B are inclusive, meaning that in most cases, only the heavy ^7Be core of the nucleus is detected, while the proton remains undetected. The present study is the first to provide an exclusive measurement of the breakup of ^8B with a precision that enables the authors to infer reliable information on both the reaction process and the structure of this exotic projectile. Moreover, as mentioned above, this study is particularly valuable because it includes a close collaboration between theorists and experimentalists. Very precise reaction calculations, which naturally account for polarisation effects and, for some of them, which include the dynamical excitation of the ^7Be core, have been performed. Multi-differential cross sections computed from these calculations have been used in the Monte-Carlo simulation of the detection setup. This is clearly one step further than most breakup measurements usually performed at radioactive-ion beam facilities. This is actually the most accurate way to confront theory and experiment.

For all these reasons, the results of this study should be quickly published. Nature Communication seems the right place to advertise this state-of-the-art measurement and its theoretical analysis, as it would interest a broad readership, from the obvious nuclear physicists, who would gain precise information about the structure of ^8B and the mechanism of reactions involving that exotic nucleus, to

astrophysicists, who could use that information to better constrain the rate of the radiative capture $7\text{Be}(p,\gamma)8\text{B}$, which is currently poorly known. Neutrino physicists could also gain from this study because a better constrained on the radiative-capture cross section would improve the reliability of the solar neutrino flux predicted by standard solar models.

I only have a few questions, which I would like the authors to answer prior to giving the green light for the publication of this work.

On the theoretical side, the authors say that the reaction calculations have been performed using the computer codes THOx and FRESKO. While the former includes dynamical core excitation, the latter, in its most usual version, does not. It is not clear in the present version of this short text, which of the calculations shown in the different figures contains core excitation. When the authors write CDCC, do they mean Fresco calculations without core excitation? Could the authors provide more precision about this, and especially emphasise when the core excitation actually matters in the calculation?

To my knowledge, the model of the core in THOx assumes a quadrupole deformation of the core, and hence, an E2 transition between the $3/2^-$ ground state and $1/2^-$ excited state of 7Be . However, the electromagnetic transition between those states seems dominated by an M1 transition. Could the authors comment on that? Does it matter in their calculation? Are magnetic transitions enabled in their reaction code?

The mechanism of the Coulomb breakup has been previously studied within multiple models of reaction. In addition to the seminal work within CDCC of J. A. Tostevin, F. M. Nunes, and I. J. Thompson, Phys. Rev. C 63, 024617 (2001), the time-dependent method has also delivered interesting aspects of the Coulomb breakup of 8B , including higher-order effects, see, e.g., H. Esbensen and G. F. Bertsch, Nucl. Phys. A 600, 37 (1996), H. Esbensen, G. F. Bertsch, and K. A. Snover, Phys. Rev. Lett. 94, 042502 (2005), and P. Capel and D. Baye, Phys. Rev. C 71, 044609 (2005). An eikonal calculation has also been able to reproduce data from various experiments, see G. Goldstein, P. Capel, and D. Baye, Phys. Rev. C 76, 024608 (2007). I find it surprising that these references have not been cited.

Would the fact that the authors' finding that the elastic breakup channel almost exhausts the 7Be yield explains why these previous studies lead to such a good agreement with experiment despite the absence of core excitation in the models?

The authors claim that their measurement is the first exclusive measurement of the breakup of 8B . I beg to differ on this. Although less precise, and at higher beam energies, other exclusive experiments have been performed in the past. The authors should cite the papers by T. Motobayashi et al., Phys. Rev. Lett. 73, 2680 (1994), T. Kikuchi et al., Phys. Lett. B 391, 261 (1997), and B. Davids et al., Phys. Rev. Lett. 86, 2750 (2001).

In Fig. 4(e), some of the data points clearly overestimate the theoretical prediction. Do these events correspond to the experimental data that overestimate the CDCC calculation in Fig. 3(b)?

The authors claim they correspond to some inadequacy of the 8B structure model. Could this instead confirm that the under-prediction of the theory at large relative energy at the beam energy 46.1MeV is due to uncertainties in the nuclear projectile-target interaction?

As mentioned before, I find this study very interesting and worth publishing. Once the authors have answered my comments, I think this manuscript will be ready for publication.

In addition to these main comments, I have also found a few typos and misprints, which I list here:

- In the authors list, I think that there shouldn't be a full stop in « La Commara ».
- line 11: « Such nuclei are »? The subject of the previous sentence are « light nuclei ».
- l.17: the elastic-scattering process is not a subsystem. Do the authors mean something like: « with the system in its ground state, e.g., during elastic scattering collisions »?
- l. 19: « influence of the continuum » (« the » missing).
- p. 2 caption of Fig. 1: « Although located in the potential well ».
- l. 46: « Despite the substantial efforts » (no « Albeit »).
- l. 116: « it is not as significant as in the » (first « as » and « in » missing).
- p. 3 In Fig. 2 it would be better to clearly identify the process to which the cross section is related: elastic scattering on the left axis using, e.g., σ_{el} and breakup on the right axis using, e.g., σ_{bu} .

In Fig. 2(a): « channel » (« a » missing).

In the caption of that same figure, two « solid curves » are mentioned. Unfortunately they are both of the same colour (red) although they correspond to different cross sections. One line is thin and the other is thick. Maybe this could be said? A different colour choice would also help.

- ll. 183 and 185: « contributions... are just minor » or « contribution... is just minor ».
- l. 187: There is no need for a full stop in the title of the section.
- l. 206 « from global parameterizations » (plural form if both p+Sn and 7Be+Sn optical potentials are derived from global parametrisations).
- l. 262: « peaks around 30° ».

Reviewer #2 (Remarks to the Author):

I have read the manuscript "Breakup of the proton halo nucleus 8B at near barrier energies". The authors have conducted an experiment that builds on earlier work on topic. The main advance on experimental side is to combine a high-efficiency setup with a facility that can produce a decent intensity of 8B particles compared to previous work to allow exclusive measurements. The authors present extensive theoretical analysis which strongly indicates the main process to be elastic breakup rather than other mechanism. This work is certainly worth publishing and probably fits in scope of Nature Communications, but could belong also any of the other more specific journals such as Physical Review Letters or Physics Letters. I leave that call for the editors.

Overall the manuscript is well written in form of a short communication or letter. However, there are few things that need to be clarified before publication in any form:

1. The authors give a good detailed background on the past work on topic. However they do not specify why they choose to use ^{120}Sn target in this case? One would think that using e.g. something like ^{64}Zn as used successfully in previous work on elastic scattering and referenced in their calculations (ref 18) would have been more logical choice. I think the choice of target should be elaborated even if it is along lines "it is what we had available at the time of this experiment". In any case, it might be useful to add some information for possible changes or dependencies of the parameters due to choice of ^{120}Sn target in the CDCC methods section or in supplementary material.

2. The excess events of breakup products around 90 degrees in center of mass frame in experimental data versus the simulation. Authors suggest that this could be due to structure model of 8B in CDCC calculations (on or about lines 278-289, fig 4). I am not sure if the further study of the underlying structure in CDCC would be beyond scope of this work to see if this is the case? Alternatively, have the authors considered that the experimental excess could be some bias in their cuts/filters on the data? If I interpret the kinematics diagram in fig 4(a) correctly, the excess for these events are at highest transverse momentum and could be because of how the data is sorted out rather than the simulation model. Or could it be something else related to the experiment, but not considered in the simulation? Target contamination, either internal (only isotopic composition of ^{120}Sn given) or some other external impurities (water, pump oil, the usual suspects)? Breakup in the PPACs before the target giving some excess not considered by the simulation? See also comment 3.

3. The experimental methods are quite thin on how the data are being processed. The question if it would be in experimental methods section or supplementary material is more of editorial choice. However, it is not clear at all what kind of cuts/filters are imposed on the data other than the given

particle identification plot. Current experienced reader can make assumptions, but it is not explicitly stated or clearly in the manuscript or supplementary material, and certainly nobody will know in future if picking up this article first time. The article would benefit a lot even from few sentences on how choosing of the ${}^7\text{Be}$ breakup products and protons are made here beyond the triggering scheme and particle identification plots. The particle identification shows as example one strip. Is this how the data was cut, or is it more of pixel by pixel from the double sided strip detectors? Are there any timing cuts for the coincidences of different telescopes to isolate the ${}^7\text{Be}+p$ system? Was timing between PPACs and Si used in any way (authors show only dE Si detector and cyclotron RF signal difference and mention that the PPACs provided particle tracks, but no mention own timing). Are there any multiplicity conditions imposed on the strip detectors? Any considerations of inter-strip events (at such close geometry at 70 mm radius for $50\times 50\text{ mm}^2$ detectors)? Again, current experienced reader can make an educated guess of detectors used and their segmentation, but this is probably not the case in the future. Also, authors refer to a setup from one lab (PPACs) and setup from another lab (Si array), but do not provide information how these were combined or arranged for this experiment. Even a simple diagram of distances in supplementary material would be a benefit for the reader. Furthermore, it is not clear if $1e4$ pps is for 8B or for total beam (out of which 20% is said to be 8B). If it is $1e4$ pps of 8B with total of $5e4$ particles going through, were there any effects observed in the PPCAs, or any issues in the triggering rate or dead-time of the system? Have the triggering efficiencies of the PPACs as well as Si detectors folded into the efficiency simulation, or is the efficiency simulation more of geometrical consideration of the system from target to the detectors rather than full setup?

4. In all relevant plots, are the experimental counts corrected for the efficiency simulation or is the simulation adjusted for the efficiency so it overlays with the experimental data?

5. In general, most of the plots would benefit a bit thicker line width for better visualization, especially if one sees this as physically printed material rather than as electronic version. Also, readers with poorer vision would benefit of slightly larger/more bold font in figures. Also color palette in fig 2 will be hard or impossible to read for color-impaired audience (use of red/green in same plot).

6. A very minor detail in Fig 2(a) label has 'one chnnel' which is probably meant to be 'one channel'.

7. The conclusions in the manuscript kind of come out of nowhere as new paragraph on line 290 as part of section that has been indicated to be "Correlations between the breakup fragments". Perhaps add here a more clear indication that the last paragraph is concluding the whole manuscript rather than the section? Or add a separate concluding section if needed.

Reviewer #3 (Remarks to the Author):

Report on Ms. NCOMMS-22-25054

Boron-8 is a fascinating nucleus under nuclear structure aspects as well as of high interest for astrophysics. e.g. as an important source of solar neutrinos. The authors of this manuscript point to another exciting property of that loosely bound system, namely resembling a femto-scale open quantum system, intimately connected to the proton halo ground state configuration of ^8B .

By measuring in coincidence the outgoing proton and ^7Be fragments, the experiment was designed to investigate the dynamics of the proton removal, aiming at clarifying whether transfer or breakup processes are the defining reaction mechanism at energies close to the Coulomb(-plus-centrifugal)-barrier. These demanding first-time coincidence measurements, relying on a secondary beam of ^8B nuclei, are a remarkable achievement of nuclear reaction physics.

The data are interpreted by state-of-the-art nuclear reaction theory applying the CDCC coupled channels method by which the coupling to unbound continuum configurations is accounted for. The results give important insights into the breakup-dynamics, revealing for the first time unambiguously the dominance of elastic breakup of the ^8B at barrier energies. Moreover, the results confirm also the features of ^8B as an open quantum systems

Beyond any doubt, the paper qualifies for publication in Nat.Comms. However before publication is recommended finally, the authors should consider and clarify the following points:

1. A widely used observable classifying the breakup of exotic nuclei is the longitudinal momentum distribution. For the sake of comparison to former data and as an indicator for the halo property of ^8B , the authors should extract and display that quantity.
2. Under nuclear structure aspects, an important question is what the data possibly reveal on the spectroscopy of ^8B . Hence, a few words on the modeling of the ^8B - ground state would be in place: Is configuration mixing considered in the CDCC calculations? What is the coupling scheme used in the calculations?
3. The model of ref.[18] has a multitude of adjustable parameters. Did the present calculations require fine tuning? If so, which were the most sensitive parameters to adjust?

4. The schematic Fig.1 should indicate also the barrier, being of importance for the removal of a p-wave proton.

5. The authors may have a look into Sham, R. et al., Nuclear breakup of B-8 in a direct fragmentation model, Phys.Rev.C 57 (1998) 2427, as an early study of the breakup reaction mechanism

6. Concerning inelastic breakup involving ${}^7\text{Be}(1/2^-)$ Cortina et al., Nuclear Physics A 720 (2003) 3 should be consulted.

Response to reviewers' comments on NCOMMS-22-25054

Dear reviewers,

Thank you for evaluating and reviewing our manuscript "Breakup of the proton halo nucleus ^8B at near barrier energies". We are especially grateful for the constructive comments and suggestions. We have considered all these comments and have performed appropriate changes to the manuscript. The comments and questions are listed point by point and marked in blue, followed by our responses.

In addition to the main manuscript, we also upload the marked-up version to make all the changes visible.

Sincerely,

Lei Yang, on behalf of all the co-authors.

Replies to reviewer #1:

We deeply thank the reviewer for acknowledging our efforts from both experimental and theoretical aspects to attack the puzzle of ^8B breakup mechanism, and for the strong words to encourage the publication of this work as well. We agree with the reviewer that the results presented in this work would interest a broad readership. We are very glad to find that the reviewer also think that "Nature Communication seems the right place to advertise this state-of-the-art measurement and its theoretical analysis".

Please refer to the below as the replies to each comment and question.

1. On the theoretical side, the authors say that the reaction calculations have been performed using the computer codes THOx and FRESCO. While the former includes dynamical core excitation, the latter, in its most usual version, does not. It is not clear in the present version of this short text, which of the calculations shown in the different figures contains core excitation. When the authors write CDCC, do they mean Fresco calculations without core excitation? Could the authors provide more precision about this, and especially emphasise when the core excitation actually matters in the calculation?

We regret that we did not make a clear statement on the CDCC calculations and caused some misunderstanding. All calculations presented in the paper correspond to standard CDCC calculations without core excitations. The THOx code is used to generate the wave functions for the bound and unbound (continuum) states of ^8B , the latter being discretized using the pseudo-state method, that is, diagonalizing the ^8B model Hamiltonian in a basis of normalizable functions, for which we use the so-called transformed harmonic oscillator (THO) basis. The THOx code is also used to compute the coupling potentials entering the set of coupled-channel equations. Finally, these coupled equations are solved using the code FRESCO.

As the reviewer mentioned, core-excitations effect could be in principle estimated by performing the extended CDCC (XCDCC). However, that requires a dedicated effort to develop

a new structure model of ${}^8\text{B}$ including both states of ${}^7\text{Be}$, and with parameters adjusted to reproduce the spectrum of ${}^8\text{B}$. This, in our opinion, goes well beyond the scope of this manuscript, where the correlations between the breakup fragments are the main topic. Although we did not include the core excitations in the calculations, the theoretical results succeeded in reproducing the angular distributions of elastic scattering and exclusive breakup. This indicates that the contribution of the core excitation might just be minor for the ${}^8\text{B}$ case. While for one-neutron halo nucleus ${}^{11}\text{Be}$ [PRL **118**, 152502 (2017)], the elastic scattering data can only be described when the core-excited admixtures are taken into account. Moreover, although the core excitations are not considered, we include the actual spin of the ${}^7\text{Be}$ core, which has been omitted in previous CDCC calculations for ${}^8\text{B}$. This allows us a more realistic model for the ${}^8\text{B}$ structure. Some details of the calculations were added in the “Methods” section of the revised manuscript.

2. To my knowledge, the model of the core in THOx assumes a quadrupole deformation of the core, and hence, an E2 transition between the 3/2- ground state and 1/2- excited state of ${}^7\text{Be}$. However, the electromagnetic transition between those states seems dominated by an M1 transition. Could the authors comment on that? Does it matter in their calculation? Are magnetic transitions enabled in their reaction code?

As explained in the previous item, we do not include the ${}^7\text{Be}$ excitation itself. Please refer to the “Methods” section in the updated manuscript.

3. The mechanism of the Coulomb breakup has been previously studied within multiple models of reaction. In addition to the seminal work within CDCC of J. A. Tostevin, F. M. Nunes, and I. J. Thompson, Phys. Rev. C **63**, 024617 (2001), the time-dependent method has also delivered interesting aspects of the Coulomb breakup of ${}^8\text{B}$, including higher-order effects, see, e.g., H. Esbensen and G. F. Bertsch, Nucl. Phys. A **600**, 37 (1996), H. Esbensen, G. F. Bertsch, and K. A. Snover, Phys. Rev. Lett. **94**, 042502 (2005), and P. Capel and D. Baye, Phys. Rev. C **71**, 044609 (2005). An eikonal calculation has also been able to reproduce data from various experiments, see G. Goldstein, P. Capel, and D. Baye, Phys. Rev. C **76**, 024608 (2007). I find it surprising that these references have not been cited. Would the fact that the authors’ finding that the elastic breakup channel almost exhausts the ${}^7\text{Be}$ yield explains why these previous studies lead to such a good agreement with experiment despite the absence of core excitation in the models?

In our former version, we only focused in the experimental and theoretical works at energies close to the Coulomb barrier. Following the reviewer’s comment, in the revised version of the manuscript we have commented also on some studies at high energies. The references mentioned by the reviewer are cited as well.

The agreement of previous theoretical calculations with experiment is indeed very likely due to the dominance of the elastic breakup component. The amount of non-elastic breakup has been estimated in some recent inclusive ${}^8\text{B}$ experiments, namely, Sparta *et al.* [PLB **820**, 136477 (2021)] and Wang *et al.* [PRC **103**, 024606 (2021)]. In all these cases (from energies around the Coulomb barrier to the much higher ones), the non-elastic breakup component, which includes the core excitation, amounts to small fraction of the total inclusive breakup of ${}^8\text{B}$.

4. The authors claim that their measurement is the first exclusive measurement of the breakup of ${}^8\text{B}$. I beg to differ on this. Although less precise, and at higher beam energies, other exclusive experiments have been performed in the past. The authors should cite the papers by T. Motobayashi et al., Phys. Rev. Lett. 73, 2680 (1994), T. Kikuchi et al., Phys. Lett. B 391, 261 (1997), and B. Davids et al., Phys. Rev. Lett. 86, 2750 (2001).

We apologize that we did not make a clear statement in the manuscript. As the reviewer pointed out, there have been several exclusive measurements of ${}^8\text{B}$ breakup at intermediate incident energies (45 – 80 MeV/nucleon) to investigate the Coulomb dissociation process. However, there are no exclusive measurements so far at energies around the Coulomb barrier. Such experiment becomes more challenging since it is not easy to carry out a coincidence measurement between the breakup fragments as was done at higher energies, due to the much reduced kinematic focusing and low-energy reaction products. The exclusive breakup measurements at lower energies offer us a great opportunity to study both the Coulomb and nuclear interaction, and the coupling effects between various reaction channels, which is one of the most intriguing topics in the field of low-energy nuclear reactions. In this sense, this work could be regarded as the first exclusive measurement at Coulomb barrier energies. We have updated the manuscript to make a more accurate statement and to include the exclusive measurements at high energies in the introduction as well.

5. In Fig. 4(e), some of the data points clearly overestimate the theoretical prediction. Do these events correspond to the experimental data that overestimate the CDCC calculation in Fig. 3(b)? The authors claim they correspond to some inadequacy of the ${}^8\text{B}$ structure model. Could this instead confirm that the under-prediction of the theory at large relative energy at the beam energy 46.1 MeV is due to uncertainties in the nuclear projectile-target interaction?

In order to facilitate comparison, we re-plot the distributions of E_{rel} and β at 46.1 MeV in Fig. 1 (a) and (b) of the present document. The correlation between E_{rel} and β is shown in Fig. 1 (c). It can be seen that CDCC-based simulations underestimate the experimental data in the range of 2-6 MeV and 60-100° for E_{rel} and β , respectively. As shown in Fig. 1 (c), events with E_{rel} of 2-6 MeV mainly locate in the region of 60-100° of β . Therefore, we agree with the reviewer’s judgement that “the events overestimate the theoretical prediction in Fig. 4 (e) correspond to the experimental data that overestimate the CDCC calculation in Fig. 3 (b)”. In this sense, we completely agree with the reviewer’s comment that the discrepancies between the data and theoretical predictions in E_{rel} and β at 46.1 MeV should share the same origin, and the most likely reason could be the uncertainties in the potential parameters of ${}^7\text{Be}+{}^{120}\text{Sn}$ and/or the simplified ${}^8\text{B}$ structure model. We have modified the corresponding statements in the manuscript accordingly.

6. In addition to these main comments, I have also found a few typos and misprints, which I list here...

We apologize for our carelessness. In the resubmitted manuscript, these typos and misprints have been revised. We really appreciate your correction. Replies to some of these comments are listed below.

FIG. 1: Distributions of (a) E_{rel} , (b) β and (c) the correlations between them at 46.1 MeV.

⇒ 1.17: the elastic-scattering process is not a subsystem. Do the authors mean something like: « with the system in its ground state, e.g., during elastic scattering collisions »?

We feel sorry that we did not make a correct statement in the manuscript. It should be the elastic-scattering “channel”, rather than the elastic-scattering “process”, that can be regarded as a subsystem. We have corrected this mistake in the revised manuscript.

Actually, we took this definition from the paper by Alexis Diaz-Torres and Antonio M. Moro, Phys. Lett. B **733**, 89 (2014) [Ref.[2] in the manuscript]. In that work, the elastic scattering is treated as a sub-system, and the continuum of breakup channels as the environment. They show that the reduction of the Coulomb-nuclear interference peak in the elastic differential cross section shows effects of quantum decoherence caused by the environment of continuum states of the ^{11}Be projectile. So we follow this concept in the present work.

⇒ p.3 In Fig. 2 it would be better to clearly identify the process to which the cross section is related: elastic scattering on the left axis using, e.g., σ_{el} and breakup on the right axis using, e.g., σ_{bu} . In the caption of that same figure, two « solid curves » are mentioned. Unfortunately they are both of the same colour (red) although they correspond to different cross sections. One line is thin and the other is thick. Maybe this could be said? A different colour choice would also help.

Following the reviewer’s comment, we have modified the Fig. 2 in the manuscript and the corresponding caption as well. Labels of the left and right axes are changed as σ_{el} and σ_{bu} . The curve of the CDCC calculation for elastic scattering is kept to be red, while that for breakup is replaced by the colour of magenta.

⇒ l. 187: There is no need for a full stop in the title of the section.

In fact, it looks unnatural also to us, but according to the Nature Communications formatting instructions, the text must be split into several specified sections like “Results”, “Discussion”, “Methods” and so on. No other section headings are permitted. Each section can be further divided into subheaded sections. We have checked some published papers, and the subheadings are written in the format of bold phrase with a full stop in the end. In any case, we need to communicate with the editor regarding this point. Thank you very much for this comment.

Replies to reviewer #2:

We acknowledge that the reviewer recognized the challenge nature of the present experiment by noting that “The main advance on experimental side is to combine a high-efficiency setup with a facility that can produce a decent intensity of ^8B particles compared to previous work to allow exclusive measurements.” To help the readers to understand the experimental procedure easily, more details about the measurement have been added to the “Methods” section of the revised manuscript. Special thanks for the reviewer’s comments.

1. The authors give a good detailed background on the past work on topic. However they do not specify why they choose to use ^{120}Sn target in this case? One would think that using e.g. something like ^{64}Zn as used successfully in previous work on elastic scattering and referenced in their calculations (ref 18) would have been more logical choice. I think the choice of target should be elaborated even if it is along lines “it is what we had available at the time of this experiment”. In any case, it might be useful to add some information for possible changes or dependencies of the parameters due to choice of ^{120}Sn target in the CDCC methods section or in supplementary material.

Thank you for the nice suggestion. In the beginning, we planned to use a high- Z target, like ^{208}Pb , which will contribute to the investigation of the Coulomb polarization effect. Due to the limitation of CRIB facility, however, the ^8B beam energy is much lower than the Coulomb barrier of the $^8\text{B}+^{208}\text{Pb}$ system, leading to the difficulties in the further studies of elastic scattering and fusion reaction. Therefore, we choose medium-mass ^{120}Sn as the target, because (i) the energy of ^8B beam that provided by CRIB could reach the region slightly above the Coulomb barrier; (ii) both Coulomb and nuclear interactions play important roles, which is helpful to investigate the Coulomb polarization, and it may also offer the opportunity to study the interplay between the Coulomb and nuclear interaction; (iii) compared with lighter system $^8\text{B}+^{64}\text{Zn}$ (Ref. [18] in the original manuscript), we expect a larger relevance of Coulomb breakup with ^{120}Sn target. The conclusions about near vs asymptotic breakup might differ for both targets indeed so it would be interesting to extend the measurements to lighter systems in order to establish a systematics about these effects. We also plan to perform experiments with high- Z targets in the future to further investigate the influence from the target. Corresponding statements have been added in the revised manuscript.

Regarding the CDCC calculations with ^{120}Sn target, it might be possible to investigate the sensitivity of the results with the interaction parameters of $^7\text{Be}+^{120}\text{Sn}$. As stated in the manuscript, due to the lack of experimental scattering data of $^7\text{Be}+^{120}\text{Sn}$ at similar energies, we needed to rely on some global parametrization potentials. In the present work, we adopted the global ^7Li potential of Cook, as it was done in the $^8\text{B}+^{64}\text{Zn}$ data analysis. We could perform some additional calculation using an alternative potential, like, Sao Paulo potential. However, since the current potential parameters reproduce the elastic scattering and exclusive breakup data properly, we did not concentrate on this sensitivity study. To make it more concise, we think it might be better not to include the sensitivity test in the manuscript.

2. The excess events of breakup products around 90 degrees in center of mass frame in experimental data versus the simulation. Authors suggest that this could be due to structure model of ^8B in CDCC calculations (on or about lines 278-289, fig 4). I am not sure if the further study of the underlying structure in CDCC would be beyond scope of this work to see if this is the case? Alternatively, have the authors considered that the experimental excess could be some bias in their cuts/filters on the data? If I interpret the kinematics diagram in fig 4(a) correctly, the excess for these events are at highest transverse momentum and could be because of how the data is sorted out rather than the simulation model. Or could it be something else related to the experiment, but not considered in the simulation? Target contamination, either internal (only isotopic composition of ^{120}Sn given) or some other external impurities (water, pump oil, the usual suspects)? Breakup in the PPACs before the target giving some excess not considered by the simulation? See also comment 3.

As we replied to Q5 of reviewer #1, the discrepancy between the experimental data and simulation in Fig. 4 (e) could be related with the uncertainties of the $^7\text{Be}+^{120}\text{Sn}$ interaction potential and/or the simplified ^8B structure model we adopted in the CDCC calculation. One item we did not include in the CDCC calculation is the core excitation. However, to evaluate the contribution from the core excitation, it requires a dedicated effort to develop a new structure model of ^8B including both states of ^7Be , and with parameters adjusted to reproduce the spectrum of ^8B . It is indeed beyond the scope of this work. On the experimental side, we have re-examined the experimental data analysis, but we did not find any obvious mistake. As we reply to the reviewer's comment 3, some details of the data analysis are explained and added in the revised manuscript.

Regarding the target, it was provided by our collaborators from Research Center for Nuclear Physics (RCNP) of Osaka University. What we know about the target material is the percentage of ^{120}Sn (97% as stated in the manuscript). Due to the target thickness (2.7 mg/cm^2) and the energy straggling of the ^8B beam, we could not identify the elastic scattering peak from other Sn isotopes in the target, and we did not observe any suspicious scattering peaks from the external contaminants of the target, as listed by the reviewer. Moreover, assuming the target is 100% ^{120}Sn , the energy calibration parameters derived from the ^8B elastic scattering are consistent with those extracted from ^7Be scattering and alpha source off-line calibration. Therefore, we did not take into account the target contaminant in the data analysis.

Regarding the breakup occurring in PPACs, actually we considered this contribution is ignorable. We installed a $\phi 20$ mm collimator just in front of the array entrance. The distances from the upstream PPAC (PPACa) to the downstream PPAC (PPACb) and to the ^{120}Sn target are 248 and 544 mm, respectively. The distance between the collimator and the target is 116 mm. Therefore, the opening angle (assuming breakup occurs in the center of PPAC) between the PPACa(b) and the collimator is 2.8° (7.2°). According the Fig. 4 (c) and (f), the opening angle between the breakup fragments, ^7Be and proton peaks around 30° . Hence the collimator has prevented almost all the breakup products in the PPACs reaching the detectors. We have added these experimental setup details in the "Methods" section of the revised manuscript. Even if the breakup fragments successfully pass through the collimator, they need to be scattered by the ^{120}Sn and then be detected by the detector. The possibility of such a two-step process is quite low.

3. The experimental methods are quite thin on how the data are being processed. The question if it would be in experimental methods section or supplementary material is more of editorial choice. However, it is not clear at all what kind of cuts/filters are imposed on the data other than the given particle identification plot. Current experienced reader can make assumptions, but it is not explicitly stated or clearly in the manuscript or supplementary material, and certainly nobody will know in future if picking up this article first time. The article would benefit a lot even from few sentences on how choosing of the ${}^7\text{Be}$ breakup products and protons are made here beyond the triggering scheme and particle identification plots.

We completely agree with the reviewer that more details of the experimental setup and data analysis should help the average reader to understand the experiment easily and clearly. We have modified the “Methods” section of the revised manuscript. The replies to the reviewer’s questions are listed below point by point.

⇒ The particle identification shows as example one strip. Is this how the data was cut, or is it more of pixel by pixel from the double sided strip detectors?

The statistics would be too low if we analyze data pixel by pixel. In practice, after the energy calibration, all the DSSD strips can be aligned in energy and hence can be treated as an ensemble. Then we can set distinct cuts for different loci of isotopes on the $\Delta E - E_r$ spectrum.

⇒ Are there any timing cuts for the coincidences of different telescopes to isolate the ${}^7\text{Be}+p$ system?

To select the ${}^7\text{Be}+p$ coincident events, we used two kinds of gates: (i) Energy gate. According to the $\Delta E - E_r$ spectrum, we set cuts for particles of ${}^7\text{Be}$ from the ${}^8\text{B}$ reaction and proton. (ii) Timing gate. To further ensure that the selected ${}^7\text{Be}$ and p are from a ${}^8\text{B}$ breakup, we also considered the timing information, then particles of different origins could be completely separated, since only the breakup products of ${}^7\text{Be}$ and p have the same TOF as the ${}^8\text{B}$ beam. For each selected breakup fragment ${}^7\text{Be}$, we search for the coincident proton within a timing window of 20 ns.

⇒ Was timing between PPACs and Si used in any way (authors show only dE Si detector and cyclotron RF signal difference and mention that the PPACs provided particle tracks, but no mention own timing).

The hitting position of the beam was determined by the time difference of the signals arriving at the ends of the x and y delay lines of PPACs cathodes. We did not use the timing difference between the PPACs and Si in the data analysis, since the distance between the upper stream PPAC and the ${}^{120}\text{Sn}$ target is quite limited (544 mm). Considering the beam energy straggling and energy loss in the target, the PPAC-target distance is not sufficient to distinguish ${}^7\text{Be}_{\text{beam}}$ and ${}^7\text{Be}_{\text{reac.}}$.

⇒ Are there any multiplicity conditions imposed on the strip detectors?

The DAQ trigger rate is not high, only about 1000 Hz, so we did not set any multiplicity conditions, that is, we recorded all the events triggered the DAQ system. The coincident events were then selected in the off-line data analysis.

Regarding the multiplicity arising from the events hitting the inter-strip gap, please refer to the reply to the following question.

FIG. 2: Setup of the detection system.

For the case that the two breakup fragments strike the same double-sided silicon detector (DSSD): in the experimental setup, the DSSD is followed by one quadrant silicon detector (QSD). Therefore, the mismatch of hitting position may occur only when the two breakup fragments strike the same section of the QSD. However, this possibility is quite low, since the opening angle between the two breakup fragments are relatively large. Actually, in the data analysis, we only found very few such kind of events (about 1 or 2). So we did not consider the multiplicity arising from these “ghost events”, which were removed from the off-line data analysis.

⇒ Any considerations of inter-strip events (at such close geometry at 70 mm radius for 50x50 mm² detectors)?

We considered the events hitting the inter-strip gap negligible. As we did not collect the event at very forward angles, the fraction of inter-strip event is not high, which is estimated to be less than 5%. Moreover, these events were rejected since we could not unambiguously determine the outgoing angle of the event hitting the strip gap, which would introduce an uncertainty around 5° in angle. The efficiency loss arising from the removing of the inter-strip events has been considered in the simulation. The cross-talk events were removed in the data analysis when we set energy cuts for ^7Be and proton in the $\Delta E-E_T$ spectra. Furthermore, we also considered the energy condition that the energy of each event recorded by the both sides of the strip detectors are consistent within 150 keV, to eliminate the inter-strip events more strictly.

⇒ Again, current experienced reader can make an educated guess of detectors used and their segmentation, but this is probably not the case in the future. Also, authors refer to a setup from one lab (PPACs) and setup from another lab (Si array), but do not provide information how these were combined or arranged for this experiment. Even a simple diagram of distances in supplementary material would be a benefit for the reader.

We have added one figure in the “Methods” section to demonstrate the combination of the PPACs and Si array. It is also shown in Fig. 2 in this document. We appreciate this nice suggestion.

⇒ Furthermore, it is not clear if 1e4 pps is for ^8B or for total beam (out of which 20% is said to be ^8B). If it is 1e4 pps of ^8B with total of 5e4 particles going through, were there

any effects observed in the PPCAs, or any issues in the triggering rate or dead-time of the system? Have the triggering efficiencies of the PPCAs as well as Si detectors folded into the efficiency simulation, or is the efficiency simulation more of geometrical consideration of the system from target to the detectors rather than full setup?

1×10^4 pps is for ^8B , rather than the total beam. We are sorry if it causes any misunderstanding, and we have modified the description in the revised manuscript, trying to make a clear statement. As stated in the manuscript, the DAQ trigger condition was the AND of the anode signal of the upstream PPAC and the OR of the silicon detectors, and we did not record all the beam particles. The typical DAQ trigger rates were about 1000 counts per second during the experiment. The efficiency of the DAQ system is higher than 90%. Therefore, the DAQ dead time could be neglected safely.

Besides the DAQ for physical runs, another independent DAQ (CRIB-DAQ) was used for the beam tuning and on-line monitoring. For this CRIB-DAQ, the down-scaled PPACa anode signal was used as the trigger, and the pileup circuit was introduced to reject the sequential signals from PPACa within 500 ns. With the data recorded by the CRIB-DAQ, we have checked that the efficiency of the PPCAs were around 95% during the beam time. For silicon detectors, we did not consider the detection efficiency loss as long as the energy of charged particle is higher than the detection threshold. Therefore, the triggering efficiencies of the PPCAs and Si detectors were not taken into account in the simulation, where only the detector geometry and detection threshold were considered.

4. In all relevant plots, are the experimental counts corrected for the efficiency simulation or is the simulation adjusted for the efficiency so it overlays with the experimental data?

We did not consider the efficiency correction for the data shown in Figs. 3 and 4. The simulation has considered the efficiency and compared with the data. We have made a clear statement on this point in the revised manuscript.

5. In general, most of the plots would benefit a bit thicker line width for better visualization, especially if one sees this as physically printed material rather than as electronic version. Also, readers with poorer vision would benefit of slightly larger/more bold font in figures. Also color palette in fig 2 will be hard or impossible to read for color-impaired audience (use of red/green in same plot).

Thank you very much for this suggestion. We have modified the figures to make a better visualization. For Fig. 2 in the manuscript, we adopted the color combination of magenta and green to show the calculation results of breakup.

6. A very minor detail in Fig 2(a) label has 'one chnnel' which is probably meant to be 'one channel'.

We are really sorry for our carelessness. This typo has been corrected in the revised version.

7. The conclusions in the manuscript kind of come out of nowhere as new paragraph on line 290 as part of section that has been indicated to be "Correlations between the breakup fragments". Perhaps add here a more clear indication that the last paragraph is concluding the whole manuscript rather than the section? Or add a separate concluding section if needed.

We agree with the reviewer on this point. However, according to the format instruction of Nature Communications, we cannot add other sections to the text except for the designated ones, like “Results”, “Discussions” and “Methods”. Therefore, the last paragraph needs to be presented as one part of the “Results and discussion” section. To make a better readability, we add a subheading “Summary” in the beginning of the last paragraph. We appreciate this nice suggestion.

Replies to reviewer #3:

We really appreciate that the reviewer gives highly complimentary remarks to the present work. We also thank the reviewer for the positive comments, which are very helpful for revising and improving the manuscript. Replies to the comments are listed below.

1. A widely used observable classifying the breakup of exotic nuclei is the longitudinal momentum distribution. For the sake of comparison to former data and as an indicator for the halo property of ${}^8\text{B}$, the authors should extract and display that quantity.

We are grateful for this suggestion. As the reviewer mentioned, the longitudinal momentum distributions of fragments from breakup reactions could provide experimental insight into the structure of the halo states. To date, studies of longitudinal momentum distributions are mainly performed with high ${}^8\text{B}$ beam energies, like, 1471 MeV/nucleon [Z. Phys. A **350**, 283 (1995)], 38 MeV/nucleon [Phys. Rev. C **54**, 1787 (1996)], 41 MeV/nucleon [Phys. Rev. Lett. **77**, 5020 (1996)], 83 MeV/nucleon [Phys. Rev. Lett. **86**, 2750 (2001)] and 36 MeV/nucleon [Phys. Rev. C **91**, 054617 (2015)]. Experimentally, measurements with high energies will benefit from the high luminosity associated with the use of thick targets and the high detection efficiency of the forward-travelling projectile-like residues. However, it becomes quite complicated when low beam energies are adopted because: (i) couplings between various reaction channels are important; (ii) Coulomb post-acceleration will affect the momentum distributions, especially for ${}^8\text{B}$, breakup of which occurs close to the target; (iii) the energy straggling and energy loss in the thick target will introduce strong uncertainties to the longitudinal momentum distribution.

In any case, we extracted the longitudinal momentum distributions of ${}^7\text{Be}$ at 46.1 MeV, which is displayed in Fig. 3 of this document. The solid curve denotes the Gaussian fitting result, and the FWHM is determined as 161 ± 25 MeV/c. The comparison with former results are listed in Table I of this document. It can be found that the result from the present work is obviously larger than those extracted from high incident energies. As discussed above, our result includes the influence from couplings, Coulomb post-acceleration and beam energy straggling. These effects, however, are still difficult to be precisely estimated. Moreover, the motivation of the present work is to investigate the breakup dynamics, rather than the structure information of ${}^8\text{B}$. Therefore, we have not included the longitudinal momentum distributions in the manuscript in order to make our main discussion more comprehensible.

2. Under nuclear structure aspects, an important question is what the data possibly reveal on the spectroscopy of ${}^8\text{B}$. Hence, a few words on the modeling of the ${}^8\text{B}$ - ground state would

FIG. 3: Longitudinal momentum distribution of ${}^7\text{Be}$ at 46.1 MeV. The solid curve represent the Gaussian fitting result.

TABLE I: Comparisons of FWHM of the longitudinal momentum distributions of fragments from the ${}^8\text{B}$ breakup reaction.

${}^8\text{B}$ energy	FWHM (MeV/c)	Ref.
46.1 MeV	161 ± 25	present work
36 MeV/nucleon	$124\pm 17^{\text{a}}$	Phys. Rev. C 91 , 054617 (2015)
36 MeV/nucleon	$92\pm 7^{\text{b}}$	Phys. Rev. C 91 , 054617 (2015)
38 MeV/nucleon	93 ± 7	Phys. Rev. C 54 , 1787 (1996)
41 MeV/nucleon	$81\pm 4^{\text{c}}$	Phys. Rev. Lett. 77 , 5020 (1996)
41 MeV/nucleon	$62\pm 3^{\text{d}}$	Phys. Rev. Lett. 77 , 5020 (1996)
1471 MeV/nucleon	81 ± 6	Z. Phys. A 350 , 283 (1995)

^a From the stripping reaction

^b From the diffraction reaction

^c On a Be target

^d On a Au target

be in place: Is configuration mixing considered in the CDCC calculations? What is the coupling scheme used in the calculations?

Because of the adopted ${}^8\text{B}$ Hamiltonian, which includes central and spin-dependent terms, the ${}^8\text{B}$ states are described with a configuration mixing. For example, the ${}^8\text{B}$ ground state is a mixture of $p1/2$ and $p3/2$ configurations coupled to the $3/2^-$ spin of ${}^7\text{Be}$. For the adopted ${}^8\text{B}$ model, the weights of the $p1/2$ and $p3/2$ configurations are 0.0589 and 0.941, respectively. These values are in reasonable agreement with ab-initio variational Monte Carlo (VMC) spectroscopic factors 0.062(1) and 0.871(4).

The coupling scheme is the following: $|l(s)j, I_c; JM\rangle$ where l is the $p-{}^7\text{Be}$ orbital angular momentum, s is the proton spin, j is the vector sum of $l + s$, I_c is the core (${}^7\text{Be}$) spin ($3/2^-$)

and J , M the total ${}^8\text{B}$ spin and its z-axis projection. This information has been added to the “Methods” section, where CDCC framework is introduced.

3. The model of ref.[18] has a multitude of adjustable parameters. Did the present calculations require fine tuning? If so, which were the most sensitive parameters to adjust?

No fine tuning was applied to the model of ref. [18]. Please note that the potential parameters were adjusted to best reproduce the known properties of the low-lying spectrum in ${}^8\text{B}$. It was not touched further for the present calculations.

4. The schematic Fig.1 should indicate also the barrier, being of importance for the removal of a p-wave proton.

Thank you for this important comment. We have modified Fig.1 in the manuscript to show the barrier structure.

5. The authors may have a look into Sham, R. et al., Nuclear breakup of B-8 in a direct fragmentation model, Phys. Rev. C 57 (1998) 2427, as an early study of the breakup reaction mechanism.

We have cited this reference in the revised manuscript.

6. Concerning inelastic breakup involving ${}^7\text{Be}(1/2^-)$ Cortina et al., Nuclear Physics A 720 (2003) 3 should be consulted.

We include this reference in the introduction as well. Thank you very much for these helpful comments.

REVIEWER COMMENTS

Reviewer #1 (Remarks to the Author):

Dear Editor,

in this resubmitted version of their manuscript, Yang et al. have clearly and satisfactorily addressed all the issues raised in my previous report. I thank them for their clear and detailed answer, as well as for their update of the manuscript. Accordingly, I see no reason to further delay the publication of this excellent work, which, as I have emphasised in my previous report, combines accurate studies in both experimental and theoretical nuclear physics.

I have only a couple of technical issues, which should be addressed by the authors prior to submission.

1. In Fig.2, the colour of the CDCC elastic-scattering cross section has been changed from red to blue. Unfortunately, the caption of that figure still mentions « red » for that curve. This should be updated.
2. In their Summary on p.6, line 322, the authors state that « the continuum has just insignificant influence on elastic scattering... » This contradicts the sentence on p.3 l. 134-135: « the influence of the continuum states on the elastic scattering cannot be neglected... » This should be amended and/or clarified. If the coupling to the breakup channel cannot be neglected, it is not insignificant.

But for those two small issues, I strongly recommend the publication of this manuscript within Nature Communications.

Reviewer #2 (Remarks to the Author):

I have read the revised manuscript and the authors' replies to all referees. I do truly appreciate the additional marked up version of the revised manuscript making it easy to see differences. I find the clarifying answers and the updates to the manuscript satisfactory. I recommend the editors to accept this manuscript for publication. I congratulate the authors for the nice result and hope to see this publication stimulate further studies on topics covered.

Reviewer #3 (Remarks to the Author):

The revised manuscript has improved considerably. However, the authors need to reconsider their decision not to touch further halo issue in this article. Beyond the general readership, the article will surely find especially large attention in the community working on exotic nuclei. A question rising immediately will be what the data are telling about the halo properties of ^8B as reflected in the longitudinal momentum distribution of the ^7Be fragment. Admitting that this aspect is not the main issue of this highly interesting work, it is worth and necessary to address it. Hence, before publication I strongly recommend to address the measured longitudinal momentum distribution in appropriate form and place in the text, without the need to show a figure.

Concerning the longitudinal momentum distribution shown as Fig.3 in the reply, I wonder about the surprisingly small error bars in view of the scattered distribution of data points. I understand that only statistical errors were included here - and in all figures of the paper as well - but a careful estimate and discussion of systematic errors would be in place before publication. This point also applies to all results mentioned and figures shown in the manuscript.

Response to reviewers' comments on NCOMMS-22-25054A

Dear reviewers,

Thank you again for evaluating and reviewing our manuscript "Breakup of the proton halo nucleus ^8B at near barrier energies". The comments and questions are listed point by point and marked in blue, followed by our responses.

In addition to the revised manuscript, we also upload the marked-up version to make all the changes visible.

Sincerely,

Lei Yang, on behalf of all the co-authors.

Replies to reviewer #1:

1. In Fig.2, the colour of the CDCC elastic-scattering cross section has been changed from red to blue. Unfortunately, the caption of that figure still mentions «red» for that curve. This should be updated.

We apologize for our carelessness. We have corrected this mistake in the revised manuscript. We really appreciate your correction.

2. In their Summary on p.6, line 322, the authors state that «the continuum has just insignificant influence on elastic scattering» This contradicts the sentence on p.3 l. 134-135: «the influence of the continuum states on the elastic scattering cannot be neglected» This should be amended and/or clarified. If the coupling to the breakup channel cannot be neglected, it is not insignificant.

We feel sorry that we did not make a clear statement in the manuscript. The sentence in the summary that "the continuum has just *insignificant* influence on elastic scattering" has been changed as "the continuum has *minor* influence on elastic scattering".

Replies to reviewer #2:

We are very glad to find that the reviewer thought all the questions are addressed clearly and properly, and we deeply thank the reviewer for recommending the publication of this work.

Replies to reviewer #3:

1. The revised manuscript has improved considerably. However, the authors need to reconsider their decision not to touch further halo issue in this article. Beyond the general readership, the article will surely find especially large attention in the community working on exotic nuclei. A question rising immediately will be what the data are telling about the halo

FIG. 1: Raw longitudinal momentum distribution of ${}^7\text{Be}$ at (a)38.7 and (b)46.1 MeV, respectively. Results with detection efficiency corrections are shown in (c) and (d), where the solid curves represent the Gaussian fitting results.

properties of ${}^8\text{B}$ as reflected in the longitudinal momentum distribution of the ${}^7\text{Be}$ fragment. Admitting that this aspect is not the main issue of this highly interesting work, it is worth and necessary to address it. Hence, before publication I strongly recommend to address the measured longitudinal momentum distribution in appropriate form and place in the text, without the need to show a figure.

Concerning the longitudinal momentum distribution shown as Fig.3 in the reply, I wonder about the surprisingly small error bars in view of the scattered distribution of data points. I understand that only statistical errors were included here - and in all figures of the paper as well - but a careful estimate and discussion of systematic errors would be in place before publication. This point also applies to all results mentioned and figures shown in the manuscript.

We completely agree with the reviewer that it is worthy and necessary to address the longitudinal momentum distribution of the ${}^7\text{Be}$ fragment. Therefore, we added the following discussions in the end of the sub-section “**Correlations between the breakup fragments**”. Meanwhile, a brief description about the longitudinal momentum distribution was

added in the **Introduction** section as well.

“Before closing this section, we would like to add a brief comment about the halo property of the ^8B nucleus as reflected in the longitudinal momentum distribution of the ^7Be fragment. Indeed, it has been found that the narrow longitudinal momentum distribution of the ^7Be fragment from breakup reactions is regarded as prominent evidence of a halo structure of ^8B . Therefore, we extracted the corresponding results from the present coincident data. Gaussian-like structures are observed at both energies, and the full widths at half maximum (FWHM) are determined as 88 ± 22 and 106 ± 14 MeV/c for 38.7 and 46.1 MeV, respectively. These results are comparable with those derived with high-energy ^8B beams, like, 93 ± 7 MeV/c at 38 MeV/nucleon [Phys. Rev. C **54**, 1787 (1996)], 85 ± 4 MeV/c at 41 MeV/nucleon [Phys. Rev. Lett. **77**, 5020 (1996)], 81 ± 6 MeV/c at 1471 MeV/nucleon [Z. Phys. A **350**, 283 (1995)], and 92 ± 7 MeV/c at 36 MeV/nucleon [Phys. Rev. C **91**, 054617 (2015)]. It is worth noting that the present result extracted from low energies includes the influence from couplings, Coulomb post-acceleration and beam energy straggling. These effects, however, are still difficult to be precisely estimated.”

Regarding the small error bars shown in the previous reply letter, we have checked that carefully and found some mistakes in the statistical error calculation. We apologize for such carelessness. The corrected longitudinal momentum distributions of ^7Be at 38.7 and 46.1 MeV are shown in Fig. 1 in this document: Fig. 1 (a) and (b) represent the raw experimental data, while results with detection corrections are shown in Fig. 1 (c) and (d). By fitting the efficiency corrected data with the Gaussian function, the FWHM is then determined. We also re-examined the calculations for the statistical uncertainties in the manuscript, and we did not find any mistake. Concerning the systematic error, it is quite challenging to evaluate it precisely. Thus we remain to show only the statistical errors in the manuscript.

REVIEWERS' COMMENTS

Reviewer #3 (Remarks to the Author):

The revised manuscript is now in the proper state and fully qualifies for publication in Nat.Comm.. The paper is recommended for publication.

Response to reviewer's comments on NCOMMS-22-25054B

Dear reviewer #3,

Thank you again for evaluating and reviewing our manuscript "Breakup of the proton halo nucleus ^8B at near barrier energies", and we deeply thank you for recommending the publication of this work.

Sincerely,

Lei Yang, on behalf of all the co-authors.